# Spontaneous Transformation of Biomedical Polymeric Silver Salt into a Nanocomposite: Physical–Chemical and Antimicrobial Properties Dramatically Depend on the Initial Preparation State

**DOI:** 10.3390/ijms231810963

**Published:** 2022-09-19

**Authors:** Klavdia A. Abzaeva, Boris G. Sukhov, Spartak S. Khutsishvili, Elena B. Tarabukina, Lev E. Zelenkov, Anna V. Nevezhina, Tat’yana V. Fadeeva

**Affiliations:** 1Laboratory of Nanoparticles, V. V. Voevodsky Institute of Chemical Kinetics and Combustion, Siberian Branch of the Russian Academy of Sciences, 630090 Novosibirsk, Russia; 2R. Agladze Institute of Inorganic Chemistry and Electrochemistry, Ivane Javakhishvili Tbilisi State University, 11 Mindeli St., 0186 Tbilisi, Georgia; 3Laboratory of Molecular Physics of Polymers, Institute of Macromolecular Compounds, Russian Academy of Sciences, 31 Bolshoi Pr. VO, 199004 St. Petersburg, Russia; 4School of Physics and Engineering, ITMO University, 49 Kronverksky Pr., Bldg. A, 197101 St. Petersburg, Russia; 5Laboratory of Cell Technologies and Regenerative Medicine, Irkutsk Scientific Center of Surgery and Traumatology, 1 Bortsov Revolutsii St., 664003 Irkutsk, Russia

**Keywords:** polyacrylic salt, silver, nanoparticles, polymer nanocomposite, plasmon–polariton-stimulated coalescence, antimicrobial activity

## Abstract

An antimicrobial polyacrylic silver salt (freshly prepared, stored for one year and model-aged) was studied by physical–chemical techniques for nanoparticle detection. In all cases, this salt represents a composite of radical-enriched macromolecules and silver(0) nanoparticles. As time passed, the initial small spherical nanoparticles were converted into larger non-spherical silver nanoparticles. The initial highly water-soluble antimicrobial solid nanocomposite almost loses its solubility in water and cannot be used as an antimicrobial agent. Unlike insoluble solid silver polyacrylate, its freshly prepared aqueous solution retains a liquid-phase consistency after one year as well as pronounced antimicrobial properties. The mechanism of these spontaneous and model-simulated processes was proposed. These results have attracted attention for officinal biomedicinal silver salts as complex radical-enriched nanocomposite substances; they also indicate contrasting effects of silver polymeric salt storing in solid and solution forms that dramatically influence antimicrobial activity.

## 1. Introduction

Bioactive polymeric metal derivatives attract special attention due to their ability to simulate many biological phenomena, act as contrast agents in MRI and their clinical use, including macromolecular complexes of radioactive metals, etc. [1,2,3,4]. The structural formula is the fundamental constant of drugs that determinates the complex of their pharmacological properties. At the same time, some works evidence that complex compounds of noble metals (as they are stated by the manufacturer) consist of metal nanoparticles by almost a half, i.e., such metal complexes do not correspond to their official structural formula (see, for example in Ref. [5]). The authors [5] explain this fact by the potential instability of the palladium compound in question, which leads to the formation of metallic nanoparticles.

Biomedical silver salts including their polymeric species are intensively studied and are used as promising multipurpose drugs (especially as antimicrobials) [6,7,8,9,10,11]. In particular, the incomplete silver salt of polyacrylic acid (PAAg) is currently proposed as a highly effective substance that exhibits a wide spectrum of biological activities [12,13,14,15], the main of them being antimicrobial [12,13]. Owing to a valuable combination of these properties, PAAg has advantages over the iron salt of polyacrylic acid—commercial medical substance Feracryl (also known as Ferasep, Haemolok, Sepgard, Sicastat, Supraheal, etc.) [16]. Recently, in biomedical studies of PAAg, there has been a rise in interest in PAAg antimicrobial properties [17]. The variety of PAAg bioactivity has broadened rapidly, for example, specific antitumor effects have recently been found [14,15]. At the same time, due to the well-known photosensitivity of silver salts, PAAg may be expected to be involved in the photo-initiated cascade reaction affording silver metal nanophases. Changes in polymers and polymer salts over time can be observed in their physical or chemical characteristics. The reasons for the changes may be, among other things, the result of interactions with the environment, for example, when oxidation leads to chain breakage. Sometimes, several age-related phenomena can act simultaneously. In general, the topic of polymer aging is vast, and its comprehensive review will require the cooperation of a large group of experts and can be an incredibly voluminous material [18]. In the framework of this article, the physical–chemical effects of PAAg aging under the influence of weathering, so-called “natural aging”, were investigated. This refers to changes in the polymer with age under the influence of several factors, primarily light radiation and ordinary temperature, including the change in the state of the metal in the polymer salt and under the influence of these factors. According to the literature data, studies of polymer nanocomposites have demonstrated a huge effect of the presence of nanoparticles on the process of physical aging [19]. However, due to the lack of consistency of work and the inconsistency of some results, the effect of inorganic nanoparticles on the aging of the polymer matrix is still difficult to predict.

Under usual daylight exposure, both colorless aqueous solutions and solid films of PAAg become weakly colored, with the color gradually turning brown. This phenomenon is likely explained by the specific optical absorption-plasmon resonance (PR) of the formed silver nanoparticles [20]. This assumption can be a subject of special study, since actual detailed phenomena of nanosilver formation could be a powerful tool for the further directional improvement of PAAg consumer features, because nanosilver exhibits unique physical–chemical and biological properties (see, for example in Refs. [6,9,20,21,22,23,24,25,26,27,28]). This information can also serve as a key to the comprehension of biomedical effects as well as to the storing behavior for PAAg and similar biomedical silver salts.

## 2. Results

### 2.1. Structure Features of Freshly Synthesized PAAg

Freshly prepared PAAg is a fine white powder. It has been found that at room temperature and daylight exposure, the colorless aqueous solution of freshly synthesized PAAg for even one hour turns pink, then yellow, and finally deep brown. The UV–Vis spectrum of this solution shows the appearance of optical absorption in the short-wave region (maximum at 270 nm) and a symmetric intensive band in the long-wave region (maximum at 409 nm); see Figure 1, black solid line 1.

The electron paramagnetic resonance (EPR) spectrum of PAAg films, freshly prepared in the dark, contains a wide (Δ*H* about 500 G) asymmetric signal with an effective *g*-factor of about 2.19–2.21, which is imposed on a narrow (Δ*H* = 8.5 G) weak singlet of Lorentzian shape with *g* = 2.0038 (Figure 2, line 1), while the starting reagents for PAAg synthesis (silver nitrate and polyacrylic acid) give no signals under these conditions. Even under daylight exposure for a few minutes, the almost colorless PAAg film and its water solution turn pink and then darken. The intensity of the broad EPR signal of solid PAAg is slightly decreased, and its shape is changed (Figure 2).

High-contrasted electrons beam spherical nanoparticles with a multimodal size distribution (from 2 to 9 nm range, average diameter of 3 ± 1 nm) are visually determined in the freshly synthesized PAAg by transmission electron microscopy (TEM); see Figure 3. According to the TEM data, the nanoparticles are spatially separated by the organic matrix at distances equal to, or larger than, their diameter. Thus, structurally, PAAg comprises a multitude of silver nanoparticles of various sizes uniformly distributed in a polymer matrix, where each particle is surrounded by a polymer shell. Similar silver nanostructures, having rather similar morphology and sizes of nanoparticles, are described in the following works [29,30,31,32]; however, the obtained PAAg is promisingly different from them in their high solubility in water.

Using the static light scattering method (SLS), the values of the weight average molecular weight M_w_ = 750,000 g/mol and second virial coefficient A_2_ = 2.6 × 10*^−^*^4^ cm^3^mol/g^2^ were obtained. The positive A_2_ indicates good thermodynamic quality of the test solution. Using the dynamic light scattering method (DLS), the scattering species of a single type were registered in water–salt solutions, the hydrodynamic size of which R_h_ decreased from 20 nm at 0.022 g/cm^3^ to 16 nm at 0.009 g/cm^3^ due to a decrease in intermolecular interactions upon dilution (Figure 4).

The value of R_h_ agrees with the obtained value of M_w_, which together with a certain thermodynamic parameter A_2_, makes it possible to conclude the molecular dispersity of the polymer in solution. It is useful to compare R_h0_ = 10 nm, which is R_h_ extrapolated to c = 0 with a radius of gyration at R_g0_ = 24 nm obtained from the SLS data. As is known, the ratio ρ = R_g0_/R_h0_ is a characteristic value for the conformation of a dissolved particle. For the sample under study, ρ = 2.4, which is close to the ρ = 2.2 theoretically calculated for polydisperse Gaussian coils in a good solution [33]. Thus, the PAAg macromolecules in the 0.15 M NaCl solution conform to the freely swollen Gaussian coils.

### 2.2. Solid Film PAAg Stored for One Year

When exposed to light at room temperature for one year, the PAAg film becomes almost black, and its watersolubility dramatically decreases. In addition, the UV–Vis spectrum of the water extracts of partially soluble PAAg film strongly changes. Thus, the optical absorption band at 200–320 nm (Figure 1, dashed line 2) significantly broadens. The PR signal of the silver nanoparticles, such as that of their classical form, almost disappears and is transformed into a broad absorption band with two implicit maxima (370 and 580 nm), gradually dropping a long-wave component up to 900 nm (Figure 1, dashed line 2).

In the EPR spectrum of the PAAg film, which was stored for one year, a broad signal (Figure 2, line 2) was detected. This signal is similar to that of freshly prepared PAAg (Figure 2, line 1), but it has a different shape and higher intensity. The EPR spectrum of the PAAg film stored for one year also contains an additional narrow signal, which is approximately twice as broad (15 G) as compared to the signal of freshly prepared polyacrylic silver salt (Figure 2, line 1). TEM images of PAAg stored for one year demonstrate a lack of initially observed small (1–10 nm) spherical silver nanoparticles and the presence of new strongly elongated nanoparticles with significantly larger size (transverse section is about 100 nm; Figure 5).

Furthermore, the process of growth and enlargement of new nanoparticles is clearly visible: it represents pole-oriented “attraction” of the initial small spherical nanoparticles on the terminal poles of the more elongated nanoparticles (Figure 5).

### 2.3. Stimulated Aging of Solid PAAg

Physical–chemical aging of a polymer and especially the polymer salt is sometimes accelerated if the temperature is increased. Equilibrium is displaced at a higher temperature, and the material may not finish in the same state as would have been achieved by a longer aging period at a lower temperature [18]. If an elevated temperature is applied to a polymer in the presence of an aggressive chemical agent (often oxygen), then this may give rise to chemical reactions that may occur only very slowly, or not at all, at ambient temperature. This is a well-studied aspect of polymer science with discussed processes during thermal aging and polymer testing procedures, for example, in the review of thermal oxidation of polymers by Professor Pospíšil and co-authors [34]. It has importance even if the polymer or its salt is not destined to be exposed to elevated temperatures during its service life. In a majority of cases, the main cause of property deterioration is photo-oxidation, which is initiated by UV irradiation [18,35], and, as a consequence, a number of photo-aging studies have been carried out to determine the weatherability of the polymers. Accelerated testing can be obtained by using UV intensities higher than those normally encountered in service.

The simulated aging processes of the solid PAAg film (photo-and thermo-induced stimulations) were monitored by the EPR technique to elucidate the reason for silver nanoparticles formation and their further evolution. The EPR monitoring of the sample carried out with even heating up to 200 °C has shown that this thermo-stimulated process is accompanied by the narrowing (from 448 to 255 G) of a broad asymmetric signal of Dysonian form that is converted into a symmetrical singlet of a Lorentzian shape. After the sample cools to room temperature, the signal is broadened again (up to 495 G), but the symmetric Lorentzian shape remains. During heating, a narrow symmetric signal with a *g*-factor of 2.0038 and width of 8 G also broadens and remains after cooling (Figure 2, lines 3 and 4).

Under UV irradiation (1 h) of the silver polyacrylate samples, the intensities of the narrow EPR signal also increase, and new additional narrow lines appear. Moreover, the broad signal is not significantly changed (Figure 6). Detection of the narrow EPR signal in the range of the magnetic field sweep 200 G (Figure 6, inset, left) under microwave saturation (*p* = 0.25–5.00 mW) and simulation have shown that it represents a superposition of four signals, three singlets, I_1_ (Δ*H* = 7.50 G; *g* = 2.0041), I_2_ (Δ*H* = 1.55 G; *g* = 2.0005), I_3_ (Δ*H* = 12.0 G; *g* = 2.0043), and one triplet, 1:2:1 (two outermost components marked with asterisk* are apart of 48.6 G) with *g* = 2.0044.

### 2.4. Aqueous PAAg Solution Stored for One Year

Noteworthy is the aqueous solution PAAgthat is also stored for one year. It is telling that in this case, the solution continues to maintain its homogeneity.

The optical properties of the aqueous PAAg solution after a year of storage also change noticeably. Its color becomes dark gray instead of brown for the aqueous solution of freshly obtained PAAg, and the solution opalescence begins to visibly appear. The optical absorption spectrum also changes and looks like “something average” between the spectrum of freshly obtained PAAg and the spectrum of aqueous drawing of the PAAg film stored for one year (Figure 1). Thus, the short-wave shoulder with a maximum at 294 nm is significantly increased (relative to the PR metal nanoparticles) compared to the freshly obtained PAAg, however, not so much as in the case of the PAAg film stored for one year. The PR itself is shifted to the long-wavelength region and has a maximum at 455 nm; however, two shoulders are also observed: shortwave at 348 nm and longwave at 560 nm.

The film formed from this solution has retained its magnetic properties (Figure 2, line 5), although both the wide and narrow signals are slightly broadened as Δ*H* = 600 G and 9.1 G, respectively. Changes in the characteristics of the EPR signals can be related to the size and shape of the magnetic nanoparticles of the nanocomposite formed, as evidenced by the data of the TEM (Figure 3). Different images of silver nanoparticles are also observed by TEM for an aqueous solution of PAAg stored for one year (compared to the stored solid PAAg films); see Figure 7. In this case, enlarged spherical or slightly elongated (ellipsoidal) silver nanoparticles are predominantly observed instead of very elongated large silver nanoparticles.

After storing an aqueous solution of PAAg for a year, a slight increase in the size of the light-scattering particles was observed at the same concentration as in a freshly prepared solution (Figure 8), which indicates conformational–structural rearrangements of both macromolecules and enlarged silver nanoparticles.

### 2.5. Antimicrobial Activity of PAAg

PAAg has pronounced antimicrobial and fungicidal activity and is promising for applications in medical practice. Results of activity (minimum inhibitory concentration (MIC), minimum bactericidal concentration (MBC) and minimum fungicidal concentration (MFC)) of the freshly synthesized PAAg water solution and the aged PAAg water solution are presented in Figure 9, Table 1 and Table 2. However, PAAg stored for one year in solid state is unsuitable for testing antimicrobial activity by conventional methods due to its practical insolubility in water.

According to microbiological studies, the freshly synthesized PAAg (Ag—8.73%) had high antimicrobial activity against various strains of Gram-negative and Gram-positive bacteria, as well as *Candida albicans* (*C. albicans*) ATCC 90028, while Gram-negative microorganisms were more sensitive than Gram-positive ones. The PAAg was characterized by MIC values for Gram-negative flora in the range from 31.25 to 62.5 μg/mL and for Gram-positive flora from 62.5 to 250 μg/mL. The MBC for Gram-negative microorganisms ranged from 31.25 to 125 μg/mL, and for Gram-positive microorganisms from 125 to 250 μg/mL. The highest MIC and MBC values for the PAAg were shown against the Gram-positive test strain *Enterococcus faecalis* (*E. faecalis*) ATCC 29212 and amounted to 250 μg/mL (MIC/MBC). The most sensitive were test cultures of microorganisms *Escherichia coli* (*E. coli*) ATCC 25922 (MIC—31.25 μg/mL, MBC—31.25 μg/mL) and *Pseudomonas aeruginos* (*P. aeruginos*) ATCC 27853 (MIC—31.25 μg/mL, MBC—31.25 μg/mL). For the reference *Staphylococcus aureus* (*S. aureus*) ATCC 25213, MBC had the same values as MIC and was 125 μg/mL. For *Klebsiella pneumoniae* (*K. pneumoniae*) ATCC 700603 and *S. aureus* ATCC 25923, MIC and MBC were 62.5 and 125 μg/mL, respectively. In addition, antifungal activity of the PAAg was also detected; thus, for *C. albicans*, MFC and MIC were determined at the same concentration of 125 μg/mL (Table 1).

Antimicrobial activity of the aged PAAg against test strains *E. coli* ATCC 25922, *P. aeruginosa* ATCC 27853, *K. pneumoniae* ATCC 700603, *S. aureus* ATCC 25213 *E. faecalis* ATCC 29212 remained unchanged (Table 2, Figure 9); for *S. aureus*, MIC increased from 62.5 to 125 μg/mL, MBC from 125 to 250 μg/mL; for *C. albicans* ATCC 90028, only MFC increased from 125 to 250 μg/mL (Table 2, Figure 9c).

## 3. Discussion

Obviously, short-wave broadened band (200–320 nm) in the optical absorption spectrum of freshly synthesized PAAg can be attributed to some superimposed and overlapping narrow bands corresponding to the carboxylic groups of the polymer [36], as well as to individual small Ag^0^ atomic clusters [37,38,39,40]. These potentially unstable silver clusters grow and coalesce to form the primary metal nanoparticles (Figure 10a). The latter begin intensively absorbing the light in the longer-wave region (symmetric band with a maximum at 409 nm) due to a specific optical phenomenon, the PR of already electroconductive silver nanoparticles (Figure 1) [20].

This indicates that the facile one-electron redox interconversion occurs in silver polyacrylate to form the primary silver metal nanophase. Thus, the reduction of Ag^+^ cations to Ag^0^ atoms interrelated to the oxidation of carboxyl anion groups in the macromolecules, giving the corresponding polyacrylate radical centers as the more likely cause of the decarboxylation processes (Figure 10a).

As seen from Figure 10a, short-wave broadened optical absorption (max 270 nm) can be caused by the contribution of light-absorbed (in this spectral region) radical centers at the transformed macromolecules of polyacrylic acid [41,42], as well as by *γ*-butyrolactone cycles [43]. The latter can be formed due to intramolecular recombination of the neighboring hetero-radical centers (Figure 10b). In this case, the polymer system must not lose its solubility in principle. Contrarily, in the case of alternative intermolecular recombination of radicals, an insoluble 3D polymer network should be formed.

In the EPR spectrum of freshly synthetized polyacrylic silver salt, a broad line can be attributed to the conduction electron spin resonance (CESR) of Ag^0^ nanoparticles formed [44,45,46]. As a rule, it considerably broadens at room temperature due to very rapid electron relaxation processes. The narrower line could be due to the complicated signals from overlapping (due to the proximity of the EPR individual parameters) both paramagnetic small nanoparticles of silver [29,47,48,49] and redox-generated carboxylate radicals in the polymer as well as radical products of its decarboxylation (Figure 10a) [50]. These assumptions on the formation of metal nanoparticles in the silver polymer salts are completely supported by TEM data: the images clearly show polydisperse silver nanoparticles of 1–10 nm in size. Polymodal distribution of the silver nanoparticles confirms the parallel processes in the solid PAAg, i.e., permanent nucleation of nanophases and a growth of the nanonuclei formed. We cannot also exclude the probable coalescence of the nanoparticles already formed that result in sharp discrete enlargement of new nanoparticles. Being stored at daylight and room temperature for one year, the PAAg film becomes almost black, and its solubility dramatically decreases. This evidences the further deep qualitative and quantitative transformations, even in the solid phase of the silver polyacrylate. Thus, the enlarged absorption band at 200–320 nm (Figure 1, dashed line 2) suggests the accumulation of light-absorbed (in this spectral region) radical centers and *γ*-butyrolactone cycles on macromolecules of the polymeric matrix [41,42,43]; see Figure 10, but not for the small silver atomic clusters, which should grow (aggregate) into larger metal nanoparticles in time. As it was described above, a significant reduction in solubility of the solid substance in one year can be explained by major intermolecular (and minor intramolecular) recombination of the radical centers to afford a cross-linked 3D polymer network.

The strong transformation of the initial PR absorption to diffuse long-wave broadening to 900 nm with two PR modes (Figure 1) is obviously the effect of subsequent growth of nanoparticle associates that becomes larger over time. In this case, the tertiary strongly coupled metallic nanostructures (its optical absorption is diffuse long-wave broadening to 900 nm) [51] may be generated from secondary enlarged elongated metal nanoparticles (its optical absorption is a complex consisting of transverse at 370 nm and longitudinal at 580 nm PR modes) obtained from primary small spherical nanoparticles (initial PR absorption at 409 nm).

An essential peculiarity of the EPR spectra of the PAAg substance after one year of aging is the increased intensity of the narrow line, likely due to overlapping of the signals from the polymer radical centers formed, i.e., carboxylate radicals (Figure 10a) [52], and the products of their further decarboxylation [50]. This fact additionally supports the above inference on the accumulation of polymeric radicals in the substance during one year. The contribution of these radicals to short-wave optical absorption with a maximum at 270 nm also sharply increases (Figure 1).

The main conclusions on the nanosilver evolution derived from the analysis of optical absorption and EPR spectra are fully confirmed by the TEM data of the PAAg film, which was stored for one year. For example, the TEM images demonstrate almost a lack of small (1–10 nm) spherical silver nanoparticles initially observed and the presence of new strongly elongated nanoparticles with greatly enlarged size (only transverse is about 100 nm, Figure 5). All the images evidence the strongly coupled metallic nanostructures (rods, wires, etc.). The elongation of novel larger nanoparticles with a high concentration of initial small spherical silver nanoparticles near the poles suggests the enlargement of the latter due to plasmon–polariton stimulation of this process. It is well known that under the action of an electric field of background, thermal or visible quanta, electrically neutral silver nanoparticles redistribute their electrons and transform into the stimulated dipole (Figure 11a). This polarization generates a strong dipole–dipole attractive force between neighboring nanoparticles, which leads to their drift toward each other to give a new doubled nanoparticle.

For this new non-spherical ellipsoid-rod, upon any further plasmon–polariton polarization, redistribution of electrical field intensity is always higher on the poles with a lower radius (effect of electric field intensity on edges). Consequently, the predominant direction of further plasmon–polariton-stimulated drift of the nanoparticles along the most electrically favorable poles, i.e., along the line of the coalescent nanoparticles enlargement, is clearly observed even on the initial coalescent nanoparticle (Figure 11b). The growth anisotropy of elongated nanoparticles in the solid phase is strongly enhanced by the absence of the rotational degree of freedom of the macromolecules and the associated nanoparticles, i.e., constant orientation of the elongated nanoparticles to the light flux and, consequently, to the vector of the electric field of the incident quanta, which will constantly produce the same direction of nanoparticle growth. Meanwhile, the counter motion of the electric dipole nanoparticles is also further enhanced by the “electric capacitor’s effect” of the oppositely charged nanoparticles separated by a nanometer layer of a dielectric, which is an organic polymer. In this case, like in any electric capacitor, the induced “electric capacitor state” can persist for a long time. An electromotive force capable of bringing together oppositely charged nanoparticles can act on the nanoparticles for the same time. This drift likely results in the formation of new enlarged nanoparticles. In addition, a drift of the remaining initial small spherical nanoparticles toward the poles of the enlarged nanoparticles is observed (Figure 5). This accumulation of initial small spherical nanoparticles at the poles of newly large elongated nanoparticles indicates the discussed mechanism of plasmon–polariton-stimulated coalescence, which is realized, not completely excluding the probable oxidative etching and Ostwald ripening mechanisms of the nanoparticles’ shape evolution.

Taking into account the dependence of signal width and the shift of g-factor on nanoparticle sizes [44,53,54], the broad line (about 500 G) of all PAAg samples (freshly synthesized, one year stored, as well as artificially exposed to photo-and thermo-aging) can be attributed to both the CESR of Ag^0^ nanoparticles [55] and to ferromagnetic properties of the silver nanoclusters [40,55,56]. According to Kawabata’s theory (see Ref. [57]), the size of the small particles is estimated by the relationship:(1)$$\Delta H=\frac{1.78\times 10^{11}\times {\left(\Delta g\right)}^{2}\times d^{2}\times \rho }{V_{F}\times M},$$
where the metal density *ρ*, the Fermi velocity *V_F_*, the peak-to-peak linewidth Δ*H* and the metal atomic mass *M* are known, while values of Δ*g*, which is the difference between the *g* value of the metal considered and the free-electron *g*-factor (2.0023), have been found by the experiments [58,59]. Thus, the particle diameters *d* of the stabilized silver-containing nanoparticles were estimated to be 10 nm (for a broad line) and about 1 nm (for a narrow line). The EPR method detects only paramagnetic nanoparticles. If the calculated results of the nanoparticles’ sizes by the Kawabata’s theory are considered to be correct, then the nanoparticles with other sizes are diamagnetic (it is well known that large-scaled metal silver is diamagnetic).

The analysis of superposition of the narrow EPR signals I_1_, I_2_, I_3_ and triplet (Figure 6, model experiments on stimulated aging of PAAg samples) shows that the latter triplet signal seems to be attributed to the carboxylate radicals 
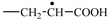
, in which the unpaired electron interacts with one *ά*-and one *β*-proton or to the radical 
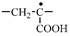
 [60,61]. Singlet I_3_ (Δ*H* = 12.00 G; *g* = 2.0043) can be assigned to radicals -CH_2_-ĊH-, which are formed via the decarboxylation reaction of polyacrylic acid [50]. Signal I_1_ can be referred to the small Ag^0^ silver nanoparticles of about 1–2 nm [48,62,63]. The singlet I_2_ (Δ*H* = 1.55 G; *g* = 2.0005) relates to acyl radicals from the polyacrylic matrix [64,65]. Moreover, the comparison of the changes in the EPR spectra upon heating and UV irradiation clearly shows that the narrow signals (*g* = 2.0038) in Figure 2, and I_1_ (*g* = 2.0043) on inset (Figure 6) are close for both processes and correspond to the nature of the narrow weak signal in the spectrum of the freshly prepared silver polyacrylate (*g* = 2.0038). Thus, such EPR singlet signal in all three cases can be attributed to the silver nanoparticles.

Thus, all radicals detected by the EPR technique can be divided into the products of the polyacrylate matrix oxidation (
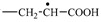
 and/or 
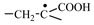
 (triplet), as well as -CH_2_-ĊH-(I_3_)) or products of this matrix reduction (
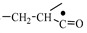
 (I_2_)). Simultaneous formation of these radicals can be rationalized by synchronous participation of the polyacrylate molecules in the redox processes on opposite poles of permanently polarized silver nanoparticles, which in this case can be considered as plasmon–polariton-stimulated electrochemical nanocells (Figure 11a). Reduction of the polyacrylate molecules can occur on the negative poles of these redox-nanocells, while oxidation can take place on the positive poles. Radical products of such a conjugated redox-process involving intensively discussed plasmon-stimulated photocatalysis on nanoparticles (see, for example in Refs. [66,67]) are here observed by the EPR method.

Thus, in a solid, highly water-soluble polyacrylic silver salt, spontaneous oxidation–reduction processes lead to a water-soluble nanocomposite of small spherical silver nanoparticles and transformed (radical-enriched) polymeric acrylate immediately after the preparation of this salt. This initial nanocomposite exhibits a complex of pronounced antimicrobial properties. Over time, in the solid phase, under the action of plasmon–polariton polarization, nanoparticles also spontaneously aggregate into large strongly elongated nanoparticles, and radical centers on neighboring macromolecules partially recombine among themselves to form a 3D polymer network. The resulting products is novel and insoluble in water and other solvents; thus, the silver-containing nanocomposite is practically unsuitable for use as a biomedical agent with antibacterial properties. On the contrary, the remaining water solubility of PAAg can be explained by the fact that in this case, there is a preferential intramolecular recombination of the radicals with the formation of intramolecular cycles, and the intermolecular crosslinking of macromolecules is not realized due to their spatial separation in solution by solvent molecules.

The optical absorption spectrum of the aged aqueous solution of PAAg, where a PR is shifted to the longwave region at 450 nm, is characteristic of large spherical nanoparticles. In addition, there are two components of another complex PR (at 348 nm and at 560 nm) that can be attributed, respectively, to the transverse and longitudinal components of PR of elongated silver nanoparticles. These PR data are also fully confirmed by the photo TEM, where both enlarged spherical and slightly elongated ellipsoidal nanoparticles are clearly observed.

Another form of enlarged silver nanoparticles (spherical and weakly extended ellipsoid nanoparticle in aged aqueous solution PAAg against the strongly elongated rod-shaped or wire-aged solid PAAg) is apparently caused by rotational mobility of macromolecules and associated nanoparticles in a solution. Firstly, it excludes the constant orientation of growing nanoparticles to the light flux (as well as to the electric field vector of the incident quanta) and prevents the highlighted direction of nanoparticles growth. Secondly, in the aqueous solution, the “electric capacitor’s effect” discussed above for the solid phase is excluded, since in this case, the water under usual conditions (without special purification, etc.) is not a dielectric but an electrical conductor.

It is obvious that the differences in the structure of the polymer matrix of the aged samples of PAAg (insoluble crosslinked intermolecular polymeric 3D nanocomposite network and water-soluble uncrosslinked nanocomposite) influence their antimicrobial properties more pronouncedly than the differences in the structure of nanoparticles. Whereas in the first nanocomposite, these properties, in general, cannot be tested by conventional methods due to its insolubility in water, the second water-soluble nanocomposite continues to demonstrate high antimicrobial properties, which are slightly reduced compared to freshly obtained PAAg (see Table 1 and Table 2) that can be explained by the coarsening of silver nanoparticles and a change in the initial structure of the polymer over time.

Undoubtedly, all the dramatic changes, observed in polyacrylate silver salt, require a further detailed investigation of not only already-known antimicrobial activities, but also of other kinds of biological activity, which is inherent both in silver nanocomposites and in radical-enriched substances.

Thus, spontaneous formation and evolution of silver nanoparticles as well as formation of polymeric radicals discovered for polymeric silver salt can serve as a key to better comprehension of its biomedical effects and aging behavior. The data obtained also could be a tool for directional improvement of features of the substance PAAg, since the nanosilver possesses diverse unusual and valuable physical–chemical and biomedical properties, and radical centers in formed silver nanocomposites should influence many radical-caused biomedical properties. This will also stimulate similar studies of other silver-based drugs, since the principal physical and chemical phenomena observed have general character. Moreover, the discussed chain growth of metal nano-wires through plasmon–polariton-induced agglomeration of the nanoparticles could be used to control nano-object growth. The data obtained should be taken into account for the storage of silver salt and nanocomposite substances for biomedical purposes; it could also be used for the development of new approaches forthe utilization of such substances in nanomedicine. In addition, such evolution processes can be used to produce immobilized silver-containing polymer coatings.

## 4. Materials and Methods

### 4.1. Materials

The PAAg was synthesized according to the published procedure in Refs. [12,15]. Acrylic acid (Sigma-Aldrich, Saint Louis, MO, USA) (9.0 g) was dissolved in water (45 mL) at room temperature under stirring on a magnetic stirrer. Then, the solution was heated to 80 °C, and potassium persulfate K_2_S_2_O_8_ (Sigma-Aldrich, Saint Louis, MO, USA) (0.045 g) dissolved in water (5 mL) was added dropwise with stirring. The resulting mixture was stirred 60 min at 80–85 °C. The polymer solution was cooled to room temperature, water (5 mL) was added, and the resulting solution was passed through a column with an anion exchanger AV-17-8 (OOO “Smoly”, Moscow, Russia). The purified aqueous solution of polyacrylic acid was poured into a thin layer and dried at room temperature for about 24 h to hard brittle transparent plates of polyacrylic acid.

The resulting polyacrylic acid (4.0 g) was dissolved in water (70 mL) at room temperature under stirring on a magnetic stirrer, and AgNO_3_(Sigma-Aldrich, Saint Louis, MO, USA) (0.53 g) in water (5 mL) was added. The solution was stirred for 30–60 min and passed through a column with an anion exchanger AV-17-8 (OOO “Smoly”, Moscow, Russia). The purified aqueous solution of PAAg was poured into a thin layer and dried at room temperature for about 24 h to hard brittle grey-brown plates of PAAg.

Silver content (8.73%) in PAAg was determined on Perkin-Elmer AAnalyst 800 (PerkinElmer, Waltham, MA, USA). Aqueous solution of freshly prepared PAAg films, the same aqueous PAAg solution (5%), stored for a year, as well as the aqueous solution (extract) of the PAAg film, stored for a year, which lost more of its water solubility over time, were also studied.

### 4.2. Physical–Chemical Measurements

The formation of silver nanoparticles within PAAg in solid state and in water solution was studied by data correlation of complementary physical–chemical methods including monitoring and simulating technique: PR optical absorption spectroscopy, TEM, SLS, DLS, as well as the already well proven for the study of biomedical nanosilver, spectroscopy of EPR including CESR in metal nanoparticles [37,68,69].

The optical absorption spectra were recorded on a Perkin Elmer Lambda 35 UV–Vis spectrophotometer for the water solution of freshly obtained and one-year-stored PAAg, as well as for the aqueous extract of a slightly soluble PAAg film stored for one year.

Freshly synthesized PAAg was studied in 0.15 M NaCl aqueous solutions by SLS and DLS methods at a constant temperature of 21 °C. The working concentration interval was *c* = (0.022 – 0.009) × 10^−2^ g/cm^3^.

Photocor FC setup with the Spectra Physic He-Ne laser (Moscow, Russia) equipped with a Photocor-FC correlator (the number of channels is 288) was used to conduct the measurements. The laser wavelength was 632.8 nm. The intensity of scattered light was measured at varying the scattering angle from 45 to 135 degrees. The Zimm theory was employed [70] to obtain a weight average molar mass M_w_, a second virial coefficient A_2_, and a radius of gyration *R*_g0_. The refractive index increment dn/dc necessary for the M_w_ calculation was measured using a KEM RA-620 refractometer (Shanghai, China), dn/dc = 0.215 cm^3^/g. The hydrodynamic radii *R*_h_ of dissolved species were obtained by the regularization method according to the DLS data. The values of *R*_h_ registered for each concentration were averaged over the scattering angles because of their angular independence.

The EPR spectra were recorded with Bruker ELEXSYS E-580 spectrometer, X-band 9.7 GHz (Billerica, MA, USA). CW EPR-spectra were recorded at the following conditions (in quartz ampoules with a diameter of 5 mm): amplitude modulation 1–10 Gs, modulation frequency 100 kHz, receiver gain 60–80 dB, time constant 0.02–0.04 s, conversion time 0.04–0.08 s, microwave power 0.6325 mW, average number of scans 20 at room temperature. To study the physical–chemical changes and supramolecular organization of the polymer salt, the aging of freshly prepared PAAg was stimulated by a single short-term treatment of the samples at 200 °C for an hour (nitrogen thermal attachment) or by UV irradiation with a broad-spectrum lamp in air atmosphere. Such methods are used to quickly screen the characteristics of materials that are subjected to accelerate artificial weathering in an atmosphere of oxygen or nitrogen [34]. The UVirradiation EPR experiments were recorded with LSB610 100W Hg lamp (UV irradiation system, ER 203 UV (Quantum Design, Darmstadt, Germany)) at room temperature. The monitoring of the thermo-stimulated process was carried out with even heating (slowly increasing the temperature by 30 degrees steps) up to 200 °C in quartz ampoules (diameter of 5 mm) directly in the resonator of EPR spectrometer. The EPR spectra simulations were performed with WINEPR SimFonia 1.26 1996 (Bruker, Billerica, MA, USA). EPR study of aqueous solutions of PAAg has not been carried out due to a high level of “parasitic” water absorption of the sounding microwave radiation of the spectrometer and a decrease due to this signal-to-noise ratio.

To study the morphology of PAAg films, the obtained salts were dissolved in water. The slightly soluble PAAg film stored for one year was cut into thin strips and extracted with water at room temperature for one day to form a colored aqueous extract above the swollen straw. Then, PAAg water solutions or aqueous extract were applied to grids with formvar supports and dried. The prepared sample films were examined using a Leo 906 E (Carl Zeiss, Jena, Germany) TEM at an accelerating voltage of 80 kV (resolution of 0.36 nm). Micrographs were taken with a MegaView II camera and processed using Mega Vision software.

### 4.3. Antimicrobial and Fungicidal Activity

Determination of antimicrobial activity of both freshly synthesized and one-year-aged PAAg water solution and their comparative assessment of MIC, MBC and MFC activity was carried out by using a serial dissolution method [71,72]. The study was conducted according to the requirements of the State Pharmacopoeia of Russian Federation. Reference museum cultures were used as test strains *E. coli* ATCC 25922, *P. aeruginosa* ATCC 27853, *K. pneumonia* ATCC 700603 (extended-spectrum *β*-lactamase), *S. aureus* ATCC 25923, *S. aureus* ATCC 25213, *E. faecalis* ATCC 29212, and *C. albicans* ATCC 90028 (Becton Dickinson, Franklin Lakes, NJ, USA). The initials solutions contained 1000 µg of PAAg in 1 mL of water. Under aseptic conditions, from these solutions, two-fold concentrations of the drug were prepared in a liquid nutrient medium with a final concentration of the microorganism of 5 × 10^5^ CFU/mL (using 8–11 tubes of 1 mL volume). Antimicrobial activity was studied at drug concentrations in the range of 0.95–500 μg/mL.

First, 18–24 h cultures of Gram-negative and Gram-positive microorganisms grown on a dense Mueller-Hinton medium were taken into experiments. For inoculation, test organisms were suspended in a concentration of 0.5 McFarland standard (1.5 × 10^8^ CFU/mL) diluted 100 times in a sterile isotonic sodium chloride solution. Bacterial suspensions were standardized using a Densi La METER densitometer (ErbaLachema, Czech Republic). To obtain the required inoculum (5 × 10^5^ CFU/mL), 50 μL of a bacterial suspension containing 10^6^ CFU/mL was added to each tube. The control tube contained 1 mL of broth without PAAg and 50 μL of culture for each tested strain. Each test was performed in duplicate. The inoculates were incubated in a normal atmosphere at 35 °C for 18–24 h. The results were evaluated visually, determining the presence or absence of growth in a medium containing a test sample with different concentrations. The last tube of the stunted row (clear broth) corresponded to the MIC of the preparation for this strain. From all transparent test tubes, seedings were made on a solid nutrient medium (Muller-Hinton agar) to determine cell viability. After incubation of crops in a thermostat (18–24 h), the lowest concentration of the drug was noted in a test tube, seeding from which did not give growth. This concentration was taken as MBC. The study of the antifungal activity of the PAAg was carried out in Sabouraud broth, following the same principle, with seeding on Sabouraud agar to determine MFC (incubation time 48 h).

## 5. Conclusions

In summary, just-obtained antimicrobial PAAg is a composite of radical-enriched macromolecules with silver(0) nanoparticles, but not pure polyacrylic silver salt, as it used to be considered. Spontaneous formation and evolution of silver nanoparticles as well as formation of polymeric radicals discovered for polymeric silver salt can serve as a key to better comprehension of its biomedical effects and aging behavior. The data obtained could also be a tool to directional improvement of features of substance PAAg, since the nanosilver possesses diverse unusual and valuable physical–chemical and biomedical properties, and radical centers in formed silver nanocomposites should influence many radical-caused biomedical properties. This will also stimulate similar studies of other silver-based drugs, since the principal physical and chemical phenomena observed have general character. Moreover, the discussed chain growth of metal nano-wires through plasmon–polariton-induced agglomeration of the nanoparticles could be used to control nano-object growth. The data obtained must be taken into account for the storage of silver salt and nanocomposite substances for biomedical purposes; they could also be used for the development of new approaches for the utilization of such substances in nanomedicine. In addition, such evolution processes can be used to produce immobilized silver-containing polymer coatings.

## Figures and Tables

**Figure 1 ijms-23-10963-f001:**
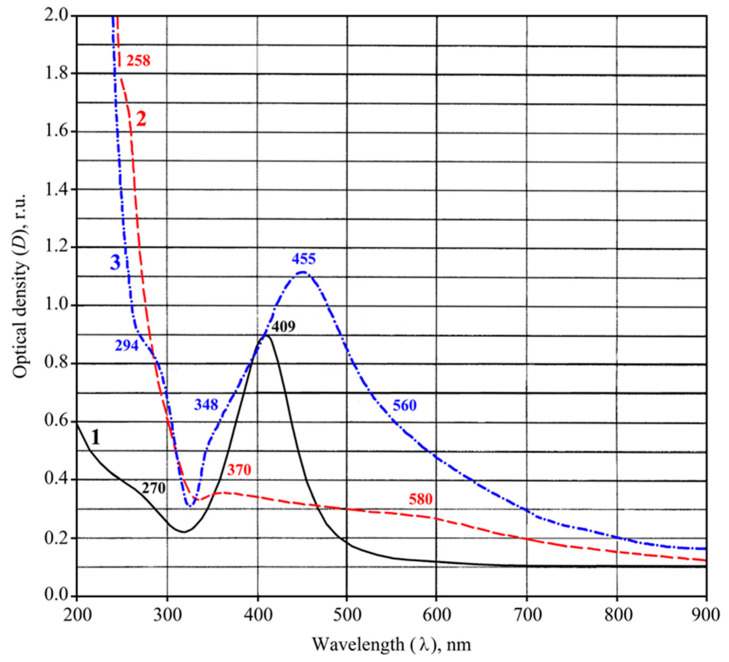
UV–Vis absorption spectra of the daylight-exposed aqueous solutions of PAAg: for one hour (black solid line 1), PAAg stored in solid state for one year (red dashed line 2), and PAAg stored as aqueous solution for one year (blue dashed line 3). In the second case, optical density was intentionally increased to show two less evident maxima of optical absorption at 370 and 580 nm.

**Figure 2 ijms-23-10963-f002:**
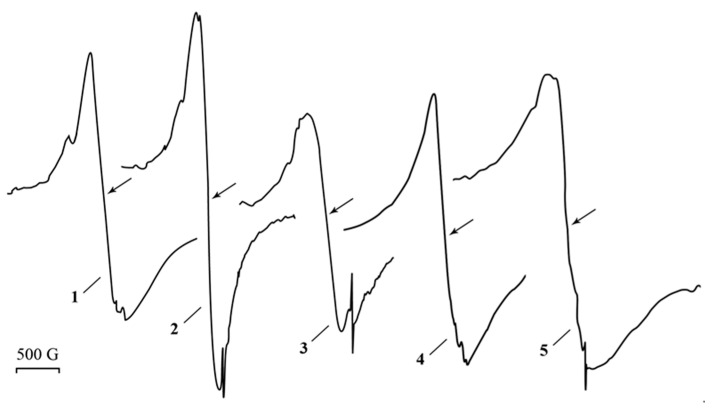
EPR spectra of PAAg: 1—freshly synthesized, 2—stored for one year; stimulated aging process of PAAg: 3—one hour after heating at 200 °C, 4—one day after heating; 5—PAAg stored as aqueous solution for one year; the arrow indicates a *g*-factor equal to 2.20.

**Figure 3 ijms-23-10963-f003:**
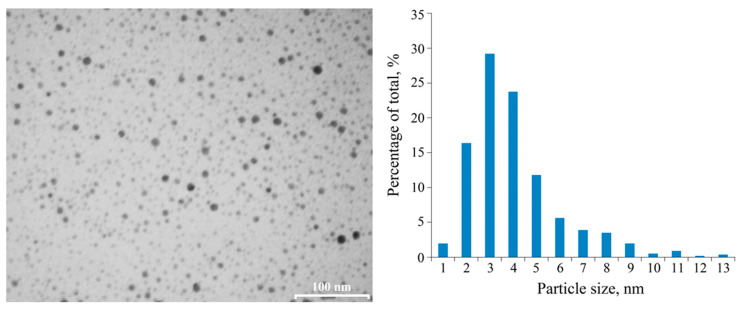
TEM image of the freshly synthesized PAAg film and size distributions of nanoparticles.

**Figure 4 ijms-23-10963-f004:**
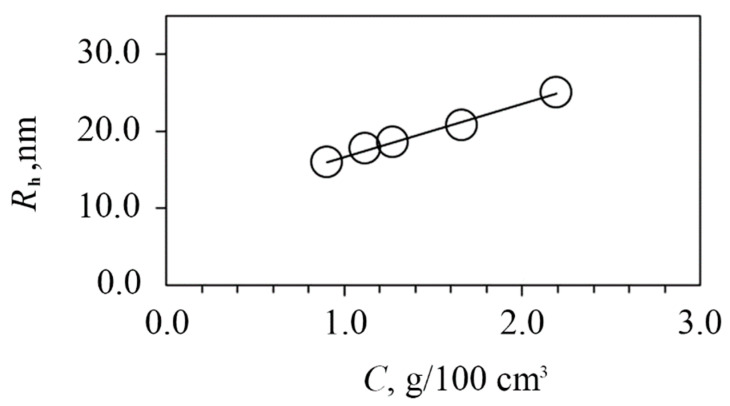
Hydrodynamic radii of dissolved species vs. concentration in 0.15 M NaCl for freshly synthesized PAAg.

**Figure 5 ijms-23-10963-f005:**
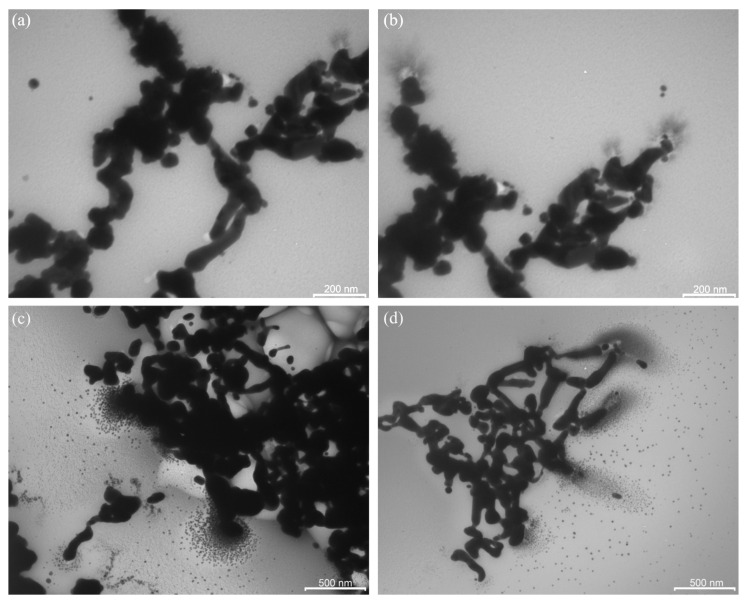
TEM images of the strongly coupled metallic nanostructures in one-year-stored PAAg; images (**a**–**d**) belong to the same sample in different scales or magnification of the image in different areas.

**Figure 6 ijms-23-10963-f006:**
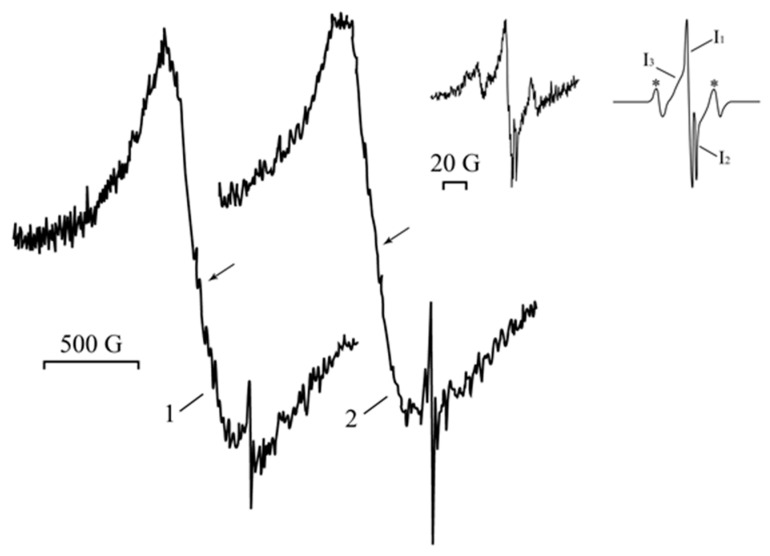
EPR spectra of PAAg film during UV-irradiation: 1—after 3 min, 2—after 1 h; inset: experimental narrow signal (**left**) and simulated (**right**), detailed explanations are below in the text; the arrow indicates a *g*-factor equal to 2.20.

**Figure 7 ijms-23-10963-f007:**
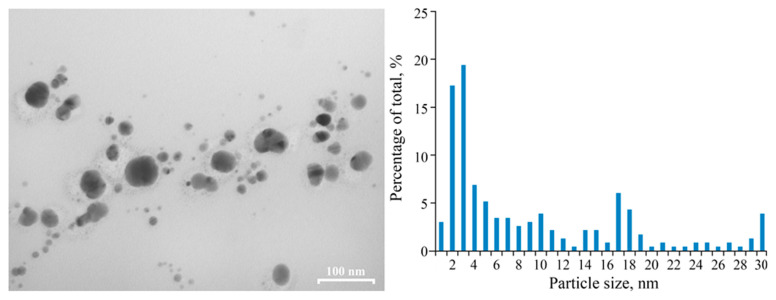
TEM image of the PAAg that was stored as aqueous solution for one year and size distributions of nanoparticles.

**Figure 8 ijms-23-10963-f008:**
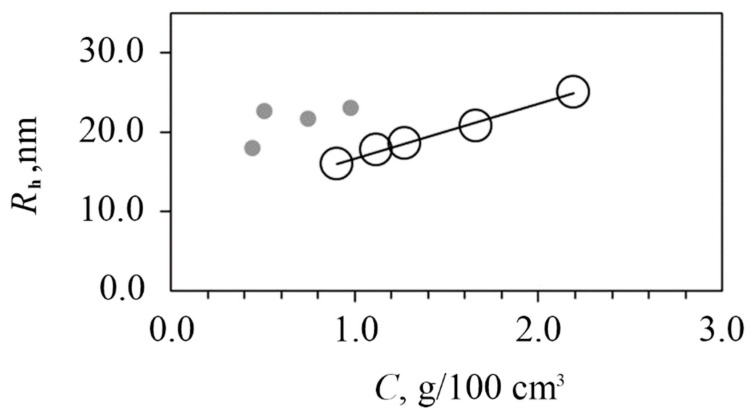
Hydrodynamic radii of dissolved species vs. concentration in 0.15 M NaCl for freshly prepared PAAg solutions (open symbols) and solutions aged after a year (grey symbols).

**Figure 9 ijms-23-10963-f009:**
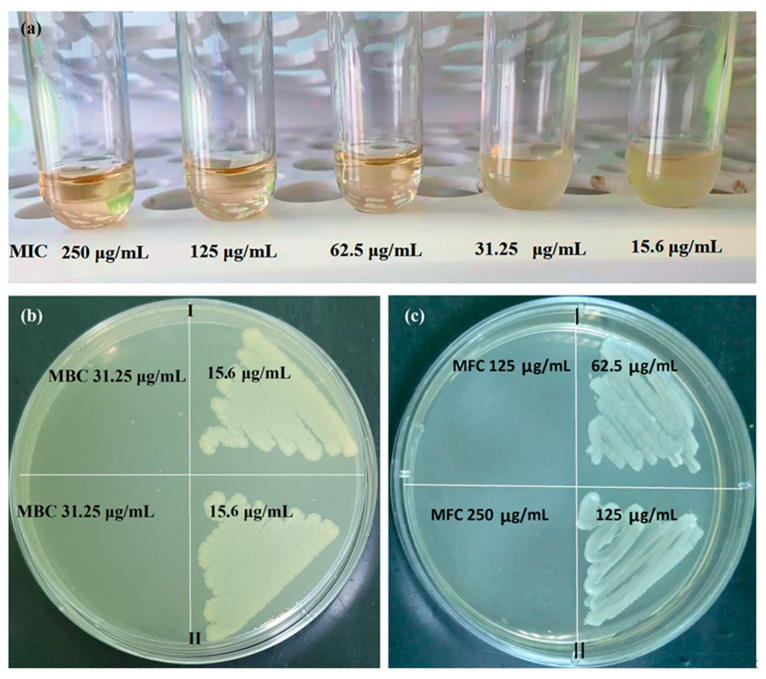
(**a**) MIC of PAAg for *K. pneumoniae* ATCC 700603 (EBSL); MBC and MFC of the water solutions freshly synthesized PAAg (I) and aged PAAg (II) for (**b**) *E. coli* ATCC 25922 and (**c**) *C. albicans* ATCC 90028, respectively.

**Figure 10 ijms-23-10963-f010:**
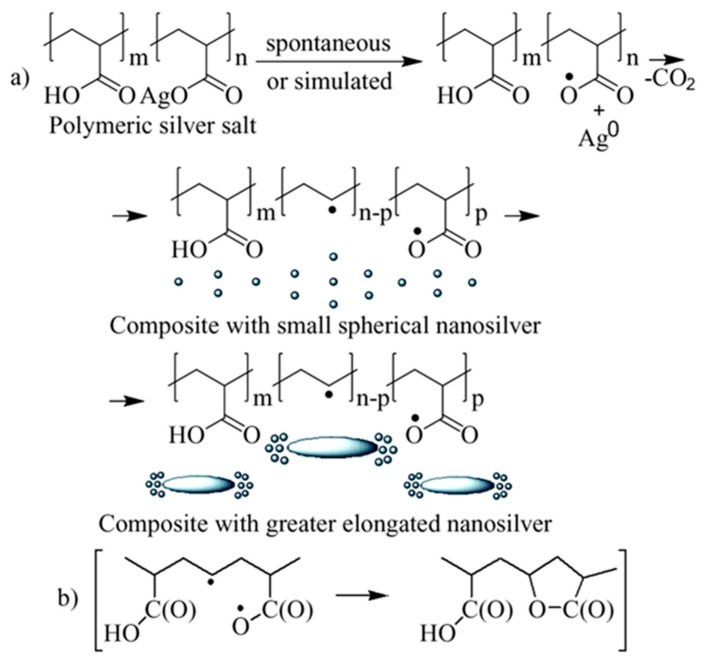
Redox mechanism of (**a**) Ag^+^ cations to Ag^0^ atoms conversion and the related polymeric radicals generation, as well as small spherical silver nanoparticles formation and their further evolution into novel and greater elongated nanoparticles; (**b**) formation of *γ*-butyrolactone cycles due to intramolecular recombination of the neighboring radical centers.

**Figure 11 ijms-23-10963-f011:**
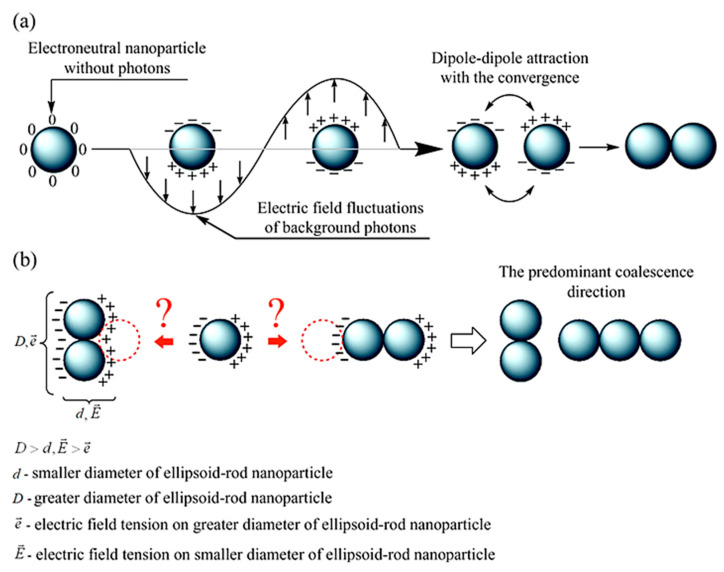
Plasmon–polariton stimulation of (**a**) nanoparticles drift toward each other; (**b**) predominant elongation of novel nanoparticles on the smallest geometrical poles, having the highest electric tension; the latter causes the most intense drift of primary spherical nanoparticles toward each other.

**Table 1 ijms-23-10963-t001:** Antimicrobial activity of freshly synthesized PAAg.

Microorganisms	MIC and MBC/MFC, μg/mL									
	500	250	125	62.5	31.25	15.6	7.8	3.9	1.9	0.95
*E. coli* ATCC 25922	−−/−−	−−/−−	−−/−−	−−/−−	−−/−−	+ +/+ +	+ +/+ +	+ +/+ +	+ +/+ +	+ +/+ +
*P. aeruginosa* ATCC 27853	−−/−−	−−/−−	−−/−−	−−/−−	−−/−−	+ +/+ +	+ +/+ +	+ +/+ +	+ +/+ +	+ +/+ +
*K. pneumoniae* ATCC 700603 (EBSL)	−−/−−	−−/−−	−−/−−	−−/+ +	+ +/+ +	+ +/+ +	+ +/+ +	+ +/+ +	+ +/+ +	+ +/+ +
*S. aureus* ATCC 25923	−−/−−	−−/−−	−−/−−	−−/+ +	+ +/+ +	+ +/+ +	+ +/+ +	+ +/+ +	+ +/+ +	+ +/+ +
*S. aureus* ATCC 25213	−−/−−	−−/−−	−−/−−	+ +/+ +	+ +/+ +	+ +/+ +	+ +/+ +	+ +/+ +	+ +/+ +	+ +/+ +
*E. faecalis* ATCC 29212	−−/−−	−−/−−	+ +/+ +	+ +/+ +	+ +/+ +	+ +/+ +	+ +/+ +	+ +/+ +	+ +/+ +	+ +/+ +
*C. albicans* ATCC 90028	−−/−−	−−/−−	−−/−−	+ +/+ +	+ +/+ +	+ +/+ +	+ +/+ +	+ +/+ +	+ +/+ +	+ +/+ +

Note: (−) Absence of test-strains growth; (+) presence of growth of test strains.

**Table 2 ijms-23-10963-t002:** Antimicrobial activity of one-year-stored PAAg water solution.

Microorganisms	MIC and MBC/MFC, μg/mL									
	500	250	125	62.5	31.25	15.6	7.8	3.9	1.9	0.95
*E. coli* ATCC 25922	−−/−−	−−/−−	−−/−−	−−/−−	−−/−−	+ +/+ +	+ +/+ +	+ +/+ +	+ +/+ +	+ +/+ +
*P. aeruginosa* ATCC 27853	−−/−−	−−/−−	−−/−−	−−/−−	−−/−−	+ +/+ +	+ +/+ +	+ +/+ +	+ +/+ +	+ +/+ +
*K. pneumoniae* ATCC 700603 (EBSL)	−−/−−	−−/−−	−−/−−	−−/+ +	+ +/+ +	+ +/+ +	+ +/+ +	+ +/+ +	+ +/+ +	+ +/+ +
*S. aureus* ATCC 25923	−−/−−	−−/−−	−−/+ +	+ +/+ +	+ +/+ +	+ +/+ +	+ +/+ +	+ +/+ +	+ +/+ +	+ +/+ +
*S. aureus* ATCC 25213	−−/−−	−−/−−	−−/−−	+ +/+ +	+ +/+ +	+ +/+ +	+ +/+ +	+ +/+ +	+ +/+ +	+ +/+ +
*E. faecalis* ATCC 29212	−−/−−	−−/−−	+ +/+ +	+ +/+ +	+ +/+ +	+ +/+ +	+ +/+ +	+ +/+ +	+ +/+ +	+ +/+ +
*C. albicans* ATCC 90028	−−/−−	−−/−−	−−/+ +	+ +/+ +	+ +/+ +	+ +/+ +	+ +/+ +	+ +/+ +	+ +/+ +	+ +/+ +

Note: (−) Absence of test-strains growth; (+) presence of growth of test strains.

## Data Availability

Data are contained within the article.

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
