# Peer review of "Spontaneous Transformation of Biomedical Polymeric Silver Salt into a Nanocomposite: Physical–Chemical and Antimicrobial Properties Dramatically Depend on the Initial Preparation State"

_ijms, 2022, doi:10.3390/ijms231810963_

Round 1

Reviewer 1 Report

Dear Authors:

I consider that the manuscript entitle 

Spontaneous Transformation of Biomedical Polymeric Silver Salt into Nanocomposite: Physical-Chemical and Antimicrobial 3 Properties Dramatic Depend on the Initial Preparation State is a very interesting contribution inside the topics of the International Journal of Molecular Sciences.

Minor corrections are suggested.

(1)   …. such metal complexes do not correspond to their official structural formula (see, for example in Ref. [5]). Please change official structural formula  to standards for structural formula….

(2)   Please plot the axes of the applied magnetic field and amplitude in figures 2 and 6.

(3)   Please explain the nanoparticles formed around the main particles, Figure 7. TEM? Has the electron beam energy of TEM some effect over the formation of smaller nanoparticles? Please check the next pictures: 

(4)   The antimicrobial features of polydisperse PAAg were well explained in the manuscript. As perspective, what would happen with the antimicrobial effect in the case on monodisperse PAAg, this would be increased?

Word file is attached.

Author Response

Dear Editor and Reviewer!

Thank you for your careful work on our article for questions and advice, which will undoubtedly help it become much better!

Further, in the course of the review, our answers will be marked in red.

Dear Authors:

I consider that the manuscript entitle 

Spontaneous Transformation of Biomedical Polymeric Silver Salt into Nanocomposite: Physical-Chemical and Antimicrobial Properties Dramatic Depend on the Initial Preparation State is a very interesting contribution inside the topics of the International Journal of Molecular Sciences.

Thank you very much for the flattering assessment of us!                                                                   

Minor corrections are suggested.

 (1)   …. such metal complexes do not correspond to their official structural formula (see, for example in Ref. [5]). Please change official structural formula  to standards for structural formula….

It is corrected.

(2)   Please plot the axes of the applied magnetic field and amplitude in figures 2 and 6.

For EPR spectra the amplitude is not very informative and is extremely rarely used, especially for such wide signals, except for very special experiments. All spectra are shown at original intensity and show well the change and scale between wide and narrow signals. In addition, for compactness and clarity, all spectra are collected side by side and it will be difficult to apply a magnetic field scale for each spectrum. However, as it should be in this case, each spectrum figure shows the scale in Gauss and the g-factor of the signal is indicated by an arrow so that it is clear in which area the signal is located. The signals are also described in detail in the text of the article. This is sufficient in such cases..

(3)   Please explain the nanoparticles formed around the main particles, Figure 7. TEM?

      In all cases, after storing the nanocomposite for one year (both in Figure 7 and Figure 5), the observation of small particles near the large particles is a frame of the dynamics of composite’s nanoparticles enlargement with time.

That is not forming small particles near the main large nanoparticles, but on the contrary, large nanoparticles are formed over time as a result of the fusion of the small nanoparticles.

This is due to the counter electrically stimulated drift of initially synthesized approximately identical small spherical metal nanoparticles, which are capable of permanent electropolarization under the action of thermal and light quanta.

We discuss such features of composite nanoparticle enlargement in the relevant sections of the article.

We discuss such features of the enlargement of the composite's nanoparticles in the corresponding sections of the article.

Has the electron beam energy of TEM some effect over the formation of smaller nanoparticles? Please check the next pictures: 

      We understand that at a certain, sufficiently high in parameters, combination of energy and electron flux density in an electron beam, stimulated ablation processes, etc. during long time, can be realized under its action. However, when observing in an electron microscope, we used routine electron energy and current density typical for conventional imaging (with much lower energy and electron flux density combination parameters than are used in electron beam ablation processes).

Therefore, there is no doubt that the electron microscope conveys to us a picture not distorted by its ray action, but a real picture in Figure 7 (and in all other Figures), which was formed in the process of the evolution of polarizable nanoparticles in the macromolecular matrix of polyacrylic acid discussed in the article.

(4)   The antimicrobial features of polydispersePAAg were well explained in the manuscript. As perspective, what would happen with the antimicrobial effect in the case on monodisperse PAAg, this would be increased?

Based on the data available in the literature, the antimicrobial activity of a fine-grained nanocomposite should be higher than that of a composite with a large grain size.

Reviewer 2 Report

This article studies the transformation of a polymer with silver nanoparticles in liquid and solid state. The presented data can be useful for various purposes.  The article es clear and the ideas are easy to understand. Though the paper is solid in the present form, there are minor comments to attend:

it would be excellent if the authors can include some TEM characterization of the fresh materials in solid state (films and solid), and for the simulated aging. This information would be valuable to complete the comparation like they did with the polymer/Silver in solution.

Details of the preparation of the solid sample for TEM characterization are missing.

There is a technical question related to the shape transformation of silver nanoparticles. Authors explained shape transformation in terms of electric effects. However, there a lot of mechanism reported for shape evolution (for instance oxidative etching, Ostwald ripening). Is it possible that other mechanism can be involved in the shape evolution of the reported nanoparticles?

Author Response

Dear Editor and Reviewer!

Thank you for your careful work on our article for questions and advice, which will undoubtedly help it become much better!

Further, in the course of the review, our answers will be marked in red.

Comments and Suggestions for Authors

This article studies the transformation of a polymer with silver nanoparticles in liquid and solid state. The presented data can be useful for various purposes.  The article es clear and the ideas are easy to understand. Though the paper is solid in the present form, there are minor comments to attend:

it would be excellent if the authors can include some TEM characterization of the fresh materials in solid state (films and solid), and for the simulated aging. This information would be valuable to complete the comparation like they did with the polymer/Silver in solution.

In the manuscript, we present a series of TEM images for freshly prepared, stored for one year in solution and in solid state of polyacrylic silver salt. However, at present, unfortunately, we do not have the corresponding TEM images for the model-aged samples in the manuscript. Also, we no longer have the same source of UV- irradiation that we used in the study to repeat the model-aging process.

Details of the preparation of the solid sample for TEM characterization are missing.

This is done on the lines 530-533 and marked in red.

There is a technical question related to the shape transformation of silver nanoparticles. Authors explained shape transformation in terms of electric effects. However, there a lot of mechanism reported for shape evolution (for instance oxidative etching, Ostwald ripening). Is it possible that other mechanism can be involved in the shape evolution of the reported nanoparticles?

The realization of these mechanisms, along with the electric plasmon-polariton-stimulated coalescence discussed in the manuscript, is not excluded completely.Therefore, the following addition was made to the text of the manuscript on the lines 365-369 “This accumulation of initial small spherical nanoparticles at the poles of new large elongated nanoparticles indicates the discussed mechanism of plasmon-polariton-stimulated coalescence, which is realized, not excluding completely the probable oxidative etching and Ostwald ripening mechanisms of nanoparticles' shape evolution.” In "keywords" also includes the expression “plasmon-polariton-stimulated coalescence”.

Reviewer 3 Report

The manuscript under review presents a comparative study of three kinds of silver-polyacrylic salt -freshly prepared, stored for one year, and stimulated model-aged by physicochemical methods, such as EPR, UV-Vis, TEM in order to evaluate the changes with the silver nanoparticles that occur during the material’s storage in different media.

The bacterial activity was studied towards large number of various strains of Gram-negative and Gram-positive bacteria. The spontaneous formation and evolution of silver nanoparticles and the formation of polymeric radicals could be a key to better comprehending biomedical effects and aging behavior.

The paper could be useful for a better understanding of the processes in pharmaceutic materials preparation.

Author Response

Comments and Suggestions for Authors

The manuscript under review presents a comparative study of three kinds of silver-polyacrylic salt -freshly prepared, stored for one year, and stimulated model-aged by physicochemical methods, such as EPR, UV-Vis, TEM in order to evaluate the changes with the silver nanoparticles that occur during the material’s storage in different media.

The bacterial activity was studied towards large number of various strains of Gram-negative and Gram-positive bacteria. The spontaneous formation and evolution of silver nanoparticles and the formation of polymeric radicals could be a key to better comprehending biomedical effects and aging behavior.

The paper could be useful for a better understanding of the processes in pharmaceutic materials preparation.

Thank you for working on our article and for rating it!
